# Prestress Modal Analysis and Optimization of Cantilever Supported Rotor under the Unbalanced Axis Force and Moving Mass

**Hao Yang \*, Jian Xu, Guoqiang Wang, Zhen Yang and Qiang Li**

College of Mechatronics Engineering, North University of China, Taiyuan 030051, China;
20050106@nuc.edu.cn (J.X.); b1701004@st.nuc.edu.cn (G.W.); yzhen@nuc.edu.cn (Z.Y.);
liqiang17@nuc.edu.cn (Q.L.)
\* Correspondence: s1901110@st.nuc.edu.cn; Tel.: +86-0351-3921-772

**Abstract:** In order to reduce the x-direction and Y-direction displacement disturbance of the barrel and improve the firing accuracy, based on Bernoulli Euler's theoretical assumption of beam and taking M134 barrel machine gun as the calculation model, the pre-stress modal analysis and optimization of cantilever supported rotor under unbalanced force and moving mass are carried out in this paper. The main work of this paper is as follows: (1) M134 physical model is established, and the unbalanced force in the motion process of projectile in bore is solved by interior ballistic theory; (2) Based on the unbalanced rotor theory, the barrel vibration model considering the projectile weight and acceleration is established; (3) The critical speed model of high-speed rotating system is established, and the critical speed is determined by finite element modal analysis to determine the rigid/flexible state of barrel components in different speed regions; (4) Based on the above model, take the x-direction and Y-direction displacement of the barrel as the output value, and take the elastic modulus of the barrel, the relative position between the barrel hoop and the fuselage components and the cross-sectional area as the variable values, carry out the optimization design, and verify the firing accuracy before and after optimization through experiments.

**Keywords:** variable structure; rotor; unbalanced axis force; moving mass; shooting effect; rounds per minute (rpm)

## 1. Introduction

Other papers have studied vibrations that are rotating but perpendicular to the axis, as paper [1,2], these aims at rotating machinery dynamics, the paper [3] aims at rotating unbalanced mass of the propeller system, and this is similar to the asymmetric force of a Gatling weapon. The paper [4–6] mainly deals with some unbalanced masses, and it is still moving. Others have studied the rotor vibrations [7,8], but the lack of component motion, and find their characteristics and better serve the motherland.

Many people study structural vibration such as a horizontal bar. If the axis is the x-axis, the force is perpendicular to the x-axis [6,9]. This paper [9] studies this kind of mechanism, which is a motor but only fixed at one end. It belongs to the unilateral motor, which is different from the common motor at both ends. In addition, there is only one tube under the force, and a mass moves along the x-axis. The rotary tube weapon studied in this paper uses a single shaft motor to drive the barrel assembly to rotate through the reducer to realize high-speed shooting. The vibration model caused by its rotation can refer to the above literature.

There is also a kind of research on rotors [10,11], but the rotor is supported on both sides. The paper [10] treats nonlinear dynamic analysis of a lightweight flexible rotor-disk-bearing system with geometric eccentricity and mass unbalance. A large deflection model has been derived to represent a nonlinear flexible rotor-bearing system to study the

bifurcation, stability, and route to chaos. The paper [11] treats novel methodology for a stochastic representation of non-linear dynamics problems. Application for estimating the stochastic non-linear responses in rotor dynamics.

Based on the above research theory, combined with the high-speed rotation characteristics of the barrel assembly of the barrel machine gun and the movement characteristics of the projectile in the bore, this paper puts forward the prestressed modal analysis and Optimization Research of the cantilever supported rotor based on the unbalanced force and moving mass, so as to reduce the displacement disturbance of the barrel in the X direction and Y direction during the shooting process of the barrel machine gun and improve the shooting accuracy.

## 2. M134 Gatling Gun

The vibration of the Gatling rotation axis is consistent with its axis, such as a rotor just with a cantilever. A Gatling gun will play a major role in short-range air defense and antimissile combat. The U.S. Army model is called the M134 Gatling gun [12–15]. The M134 machine gun is a 6-barrel aircraft gun developed by General Electric Company of the United States. It is mainly equipped on helicopters and can also be used as a vehicle weapon for mechanized infantry. It is to kill and assemble living targets and air defense. The M134 Gatling gun is composed of several barrels arranged in a ring, as Figure 1.

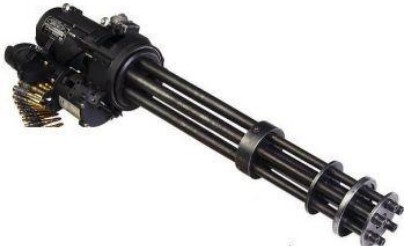

**Figure 1.** M134 Gatling gun.

In this paper, the critical speed model of high-speed rotating system is established, and the critical speed is determined through finite element modal analysis to determine the rigid/flexible state of barrel components in different speed regions; based on trajectory theory and unbalanced rotor theory, the barrel vibration model considering projectile weight and acceleration is established, and the muzzle is determined through the prestressed modal analysis of cantilever supported rotor under unbalanced force and moving mass Firing accuracy of vibrating counter rotating tube machine gun.

Each barrel rotates rapidly around the rotation center axis and reaches the stationary firing position in turn. Although the high-speed rotating barrel will increase the firing dispersion due to the centrifugal force, high firing speed and strong firepower can make up for the lack of accuracy, which makes M134 a very effective weapon to kill the living targets of the group.

The barrel and machine support of the Gatling, reference paper [16] M134 imitation barrels and body frame. Since the barrel rotates at high speed, the solution of vibration characteristics of a Gatling gun is very complicated, as shown in Figure 2.

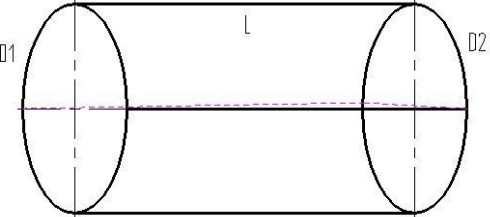

**Figure 2.** Gatling gun's barrels.

When the front and rear diameters $D1$ and $D2$ of the Gatling weapon are not equal, the $\alpha$ is equal to,

$$\alpha = \arctan\left|\frac{D1 - D2}{2L}\right| \tag{1}$$

when the front and rear diameters $D1$ and $D2$ of the Gatling weapon are equal, the angle $\alpha$ s equal 0.

Coriolis force $F_k$ is showing as:

$$F_k = -2m_d\omega v_0 \tag{2}$$

Substituting the angle, it is convenient to calculate:

$$F_k = 2m_d\omega v_0 \sin(\alpha) \tag{3}$$

If $\alpha$ t is equal to 0, then the Coriolis force $F_k$ and the moment will be equal to 0. According to the internal ballistic equation, the $p$-value can be determined:

$$SP(l + l_\psi) = f\omega\psi - \frac{\theta}{2}\varphi m v^2 \tag{4}$$

The pressure P and velocity v are obtained separately as shown in Figure 3:

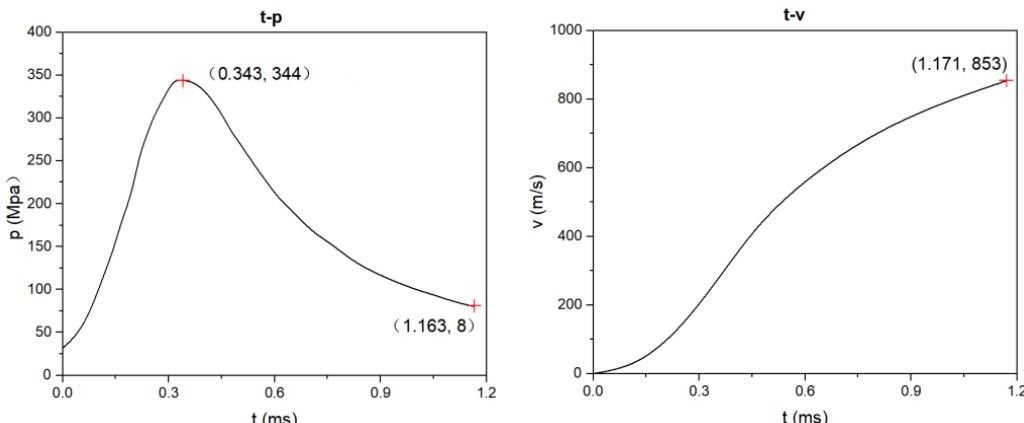

**Figure 3.** Gatling ballistic curve.

When the 7.62 × 51 mm NATO cartridge moves in the barrel, under the joint action of gunpowder gas and rifling, the projectile moves linearly along the barrel axis on the one hand and rotates on the other hand. At this time, there is a certain force on the bullet and barrel. From Equations (5)–(8), the rifling guide side will give the bullet a driving edge force N and give the barrel's lateral force and axis force. As shown in Figures 4 and 5, the OZ axis is parallel to the barrel axis, and the Z axis direction is the bullet movement direction. N is driving edge force N, fN is friction force. The lateral force on the barrel is $F_1$ and the axial force is $F_2$.

$$N = \frac{1}{n}\left(\frac{\rho}{r}\right)^2 \frac{SP_d tg\alpha + \varphi_1 K_a m v^2}{\varphi_1(\cos\alpha - f\sin\alpha)} \tag{5}$$

The circumferential torque is:

$$M = n \cdot r \cdot N(\cos\alpha - f\sin\alpha) \tag{6}$$

The lateral force of the barrel is:

$$F_1 = -n \cdot N(f\cos\alpha + \sin\alpha) \tag{7}$$

The axis force of the barrel is:

$$F_2 = n \cdot N(\cos\alpha - f\sin\alpha) \tag{8}$$

when: $n$ is the number of rifling elements, from Table 1, $n = 4$; $r$ is the warhead radius; $N$ is the driving force; $f$ is the friction coefficient, and the friction coefficient between metals, generally, it varies from 0.16 to 0.20, $f = 0.18$.

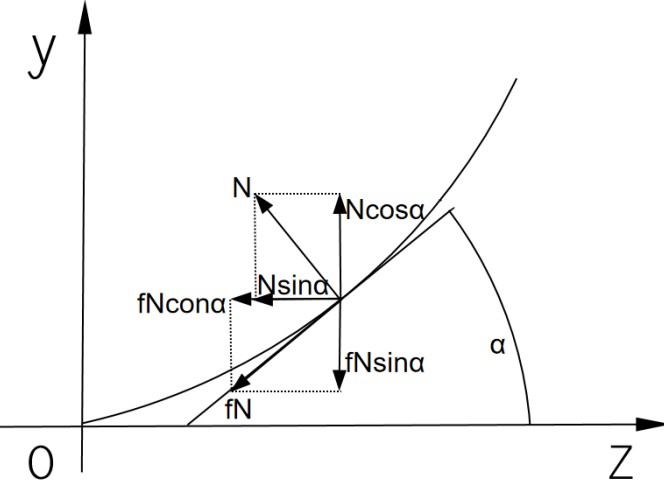

**Figure 4.** Driving edge force N and friction force fN.

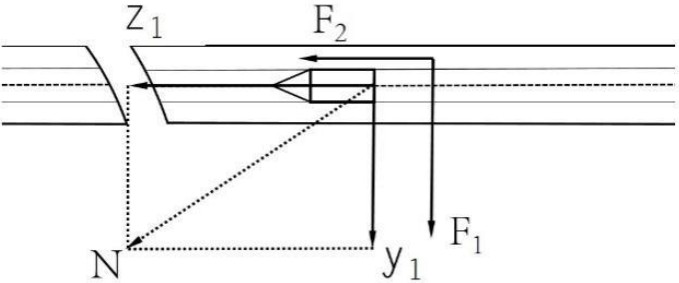

**Figure 5.** Lateral force $F_1$ and axis force $F_2$.

**Table 1.** Data of M134 [13–15] are shown.

| M134 Gatling Gun Data Parameters | Value |
|---|---|
| Whole gun weight | 15.9 kg |
| Gun length | 801.6 mm |
| Barrel length | 559 mm |
| Tube rifling lines | 4 |
| Tube rifling lines direction | Right |
| Wrapping distance | 254 mm |
| Warhead Quality | 9.75 g |
| Initial velocity | 838 m/s |
| Maximum pressure | 345 MPa |
| Effective Range | 800 m |
| Stray bullet Range | 5000 m |
| MRBF | 250,000 r |
| Lifetime | 600,000 r |
| Firing-rate | 300 rpm (speed of DC motor); 2000 rpm (practical firing rate); 6000 rpm (maximum firing rate) |
| Error | 800 m 0.2–0.8 m; 5000 m 1.5–3 m |

The $N$, $M$ is calculated as showing Figures 6 and 7:

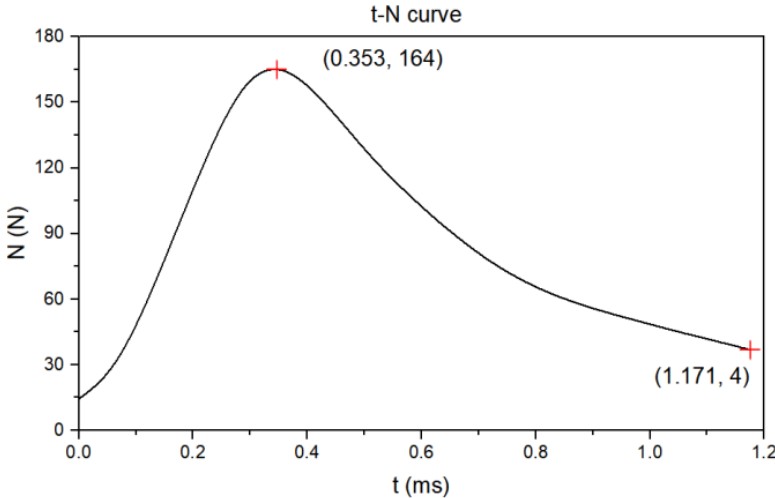

**Figure 6.** Value of N.

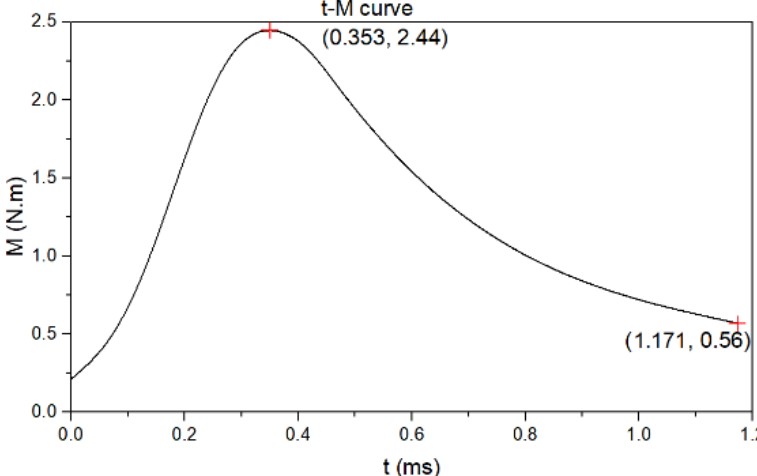

**Figure 7.** Value of M.

The pressure of internal propellant gas belongs to the impact type of external force. The force on each rifling $F_1$ and the force on each rifling $F_2$ are shown in Figure 8.

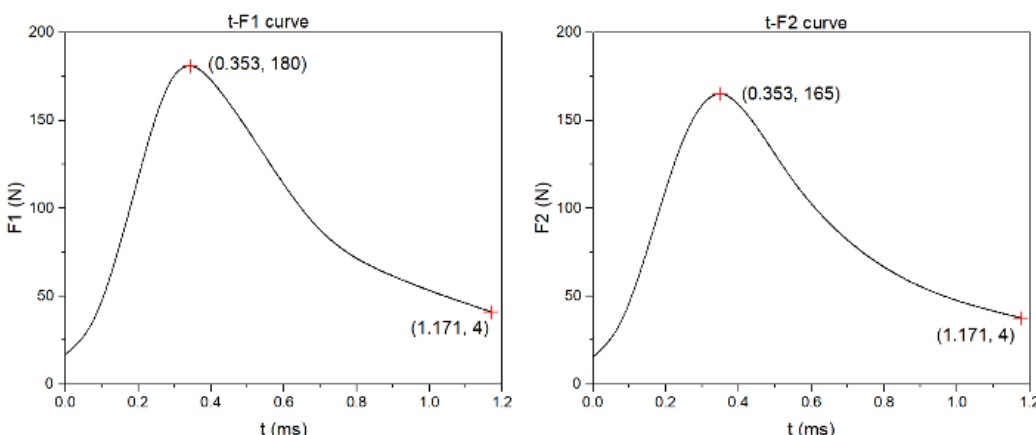

**Figure 8.** Lateral force $F_1$ and axial force $F_2$.

### 3. Theoretical Model

Depending on the hypothesis of the Bernoulli-Euler beam theory [7,17–22], the tube vibration function with moving force and moving mass was solved [10,23–25]. Therefore, two theories are used, and the first is the ballistic theory, the second is the unbalanced rotor theory.

The vertical deformation of the barrel axis is represented by $y_1$, and the longitudinal deformation of the barrel axis is represented by $y_2$, as shown Figure 5, The vibration equation of the barrel considering the weight and acceleration of the projectile is:

$$\frac{\partial^2}{\partial x^2}[EI(x)\frac{\partial^2 y_1(x,t)}{\partial x^2}] + c_1\frac{\partial y_1(x,t)}{\partial t} + \rho(x)A\frac{\partial^2 y_1(x,t)}{\partial t^2} = -(mS + F_2)\delta(x-\xi) \quad (9)$$

$$\frac{\partial^2}{\partial x^2}[EI(x)\frac{\partial^2 y_2(x,t)}{\partial x^2}] + c_2\frac{\partial y_2(x,t)}{\partial t} + \rho(x)A\frac{\partial^2 y_2(x,t)}{\partial t^2} = (mg + F_1)\delta(x-\xi) \quad (10)$$

$$S = \frac{\partial^2 y_1}{\partial t^2} + 2v\frac{\partial^2 y_1}{\partial x\partial t} + v^2\frac{\partial^2 y_1}{\partial x^2} \quad (11)$$

$$\zeta = \int_0^t v(t)dt \quad (12)$$

$$y = \sqrt{y_1^2 + y_2^2} \quad (13)$$

where: $E$ is the elastic modulus of the material; $I(x)$ is the moment of inertia of the cross-section of a cage-like structure; $c_1, c_2$ is the viscous damping coefficient; $y_1$ is the vertical deformation of the axis; $y_2$ is the longitudinal deformation; $y$ is the resultant displacement. $\rho(x)$ is the material density; $A$ is the cross-sectional area of the barrel; $I(x)$ and $A$ are constants under the simplification of uniform section; $m$ is the mass of the projectile; $F$ is the axial force of barrel from muzzle to tail, $v(t)$ is the moving speed of the projectile; $g$ is the acceleration of gravity; $\xi$ is the travel of the bullet from interior ballistic calculation; $S$ is the additional term of an external force acting on the barrel after considering the inertial effect of the bullet, and $\delta(\ldots)$ is the Dirac function.

Similar to as Figure 9, the calculation formula of $I(x)$ is as follows:

$$I(x) = nm_{sg}(\frac{D}{2})^2 + I_{zg} \quad (14)$$

where: $I_{zg}$ is the moment of the barrel connector and 6 barrels, $m_{sg}$ is the mass of the bullet, $n$ is the number of the bullet. After calculation, its internal ballistic time is only 1.2 ms. The practical firing rate of M134 is 2000 rpm, that is, $n = 1$;

$$I(x) = m_{sg}(\frac{D}{2})^2 + I_{zg} \quad (15)$$

The bullet goes through an unbalanced force, under the action of the unbalanced force, the rotor will produce vibration. At high speed, even a small mass eccentricity will produce a larger centrifugal force. The data are obtained from their modeling, and some are obtained from relevant literature [13–15,17].

The gun has an rpm from 300 to 6000, and the time interval is 200 ms to 10 ms. By the time the second bullet was started, the first bullet was already out of the gun. There is no interaction between the first and second bullet. In addition, two bullets do not appear in the barrels at the same time. The Gatling gun rotation rate is at Table 2.

According to the size of each part, the solid is drawn by 3D software, and then imported into the finite element software, which is divided by hexahedral mesh, and the calculated model is obtained.

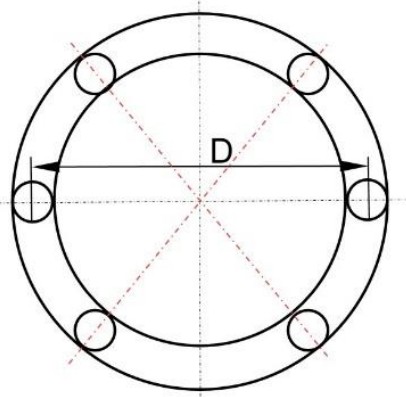

**Figure 9.** The position of the bullet in the weapon.

**Table 2.** Gatling gun rotation rate.

| Gatling Gun rpm | Speed of Revolution Round/s | Rotor Speed (rad/s) |
|---|---|---|
| 300 | 0.83 | 5.21 |
| 500 | 1.38 | 8.66 |
| 1000 | 2.77 | 17.39 |
| 2000 | 5.55 | 34.9 |
| 3000 | 8.33 | 52.3 |
| 4000 | 11.11 | 69.77 |
| 5000 | 13.88 | 87.17 |
| 6000 | 16.66 | 104.62 |

During the rotation of the barrel group continuously driven by the motor, the bullet moves along the barrel from back to front, which will cause the vibration caused by the additional mass. In addition, it is also a recognition that the bullet causes the additional vibration of the barrel assembly due to the unbalanced mass. Therefore, it is better that the lighter mass of the bullet and which will not affect the barrel assembly.

$$I_{zg}\ddot{\theta} = m_{sg}\left(\frac{D}{2}\right)^2 \omega_{zg}^2 \tag{16}$$

The above formula shows that once there is $\omega_{zg}$, the acceleration will change, which aggravates the change of $\omega_{zg}$ and aggravates the speed fluctuation.

Simultaneously, there is a discovery that there is no distinction of the energy but the speed.

## 4. Distinction between Rigidity and Flexibility of Cantilever Rotor

The cantilever rotor was studied, and the axis coincides with the cantilever beam, the XY plane is connected with other structures, and the structure rotates around the Z axis, as Figure 10. It calls a rotary squirrel cage cantilever [3].

Hence, it is necessary to study the influence of the bullet launching process of Gatling weapon, especially the force and moment are caused when the bullet moves in the barrel. If the practical firing rate is 2000 rpm, then the rotation rate $\omega$ is 34.9 rad/s.

In the rotating system, the mass center of each micro section of the rotor can not be strictly on the rotating axis. Therefore, when the rotor rotates, there will be lateral interference, and it will cause strong vibration of the system at some speed. In this case, the speed is the critical speed [26,27].

The purpose of understanding the critical speed is to try to make the compressor's working speed avoid the critical speed, to avoid resonance. Generally, the rated working speed n of the centrifugal compressor shaft is either lower than the first critical speed $n_1$ of the rotor, or between the first critical speed $n_1$.

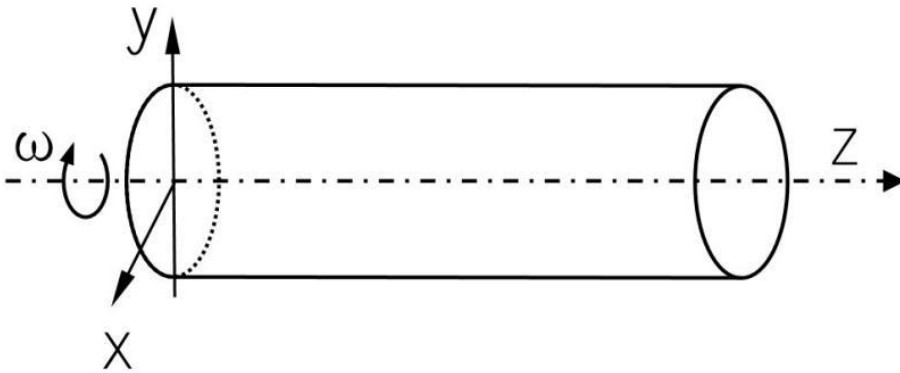

**Figure 10.** Rotary cantilever structure with Z axis.

Let's give it a modal analysis, add it some rotation velocity of 34.9 rad/s, it gives me some results, let us see that it is a component: a rigid or flexible component comes from at a certain speed.

$$\text{Rigid requirements}: \ n \ \leq \ 0.7n_1 \tag{17}$$

$$\text{Flexible requirements}: \ n \ \geq \ 1.3n_1 \tag{18}$$

This is the Campbell diagram for the M134 Gatling gun, as Figure 11.

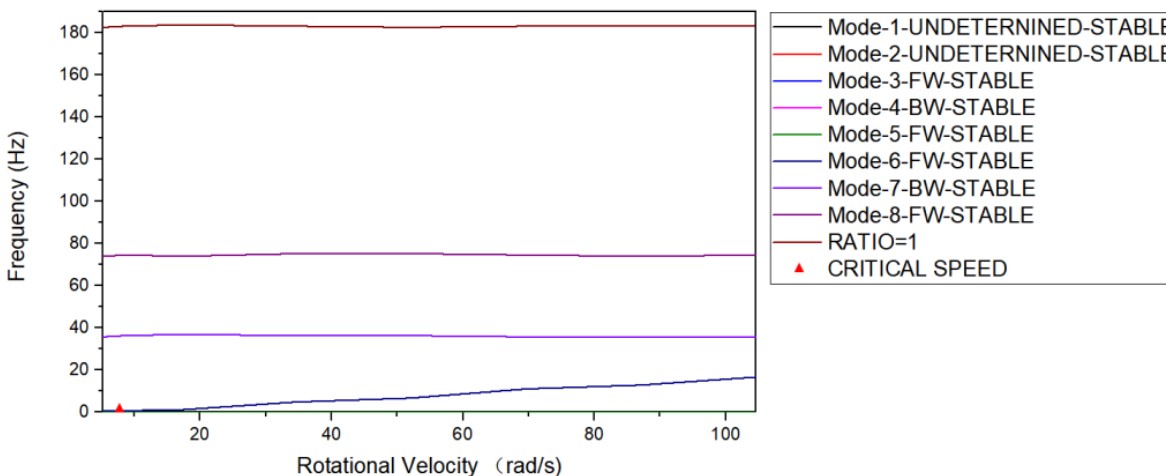

**Figure 11.** Campbell diagrams for M134 Gatling gun.

From Figure 11, critical speed $n_1$ = 8.0248 rad/s, from the Table 2, Gatling gun rpm is 300–500, this M134 is rigid rotor; and for others, rpm is 1000–6000, the M134 is the flexible rotor. M134 usually works under a flexible barrel. With or without rotation rate, the M134 Gatling gun has the first 8 frequencies, as shown in Table 3 below:

**Table 3.** With or without rotation rate, M134 fundamental frequency.

| Mode | No Rotation Rate/HZ | Used Rotation Rate/HZ | Prestress (Rotation Rate)/HZ |
|------|---------------------|-----------------------|------------------------------|
| 1 | 0 | 0 | 0 |
| 2 | 36.056 | 36.056 | 0 |
| 3 | 36.075 | 36.075 | 28.281 |
| 4 | 88.572 | 88.572 | 51.439 |
| 5 | 193.15 | 193.07 | 51.76 |
| 6 | 193.34 | 193.43 | 105.72 |
| 7 | 329.63 | 329.63 | 178.27 |
| 8 | 794.33 | 794.33 | 178.31 |

As Prestress the mode is shown as Figure 12.

However, under the action of tension force, the structural stiffness will increase and the frequency will increase. On the contrary, under the action of compression force, the structural stiffness will decrease and the frequency will decrease. This prestress is a compression force.

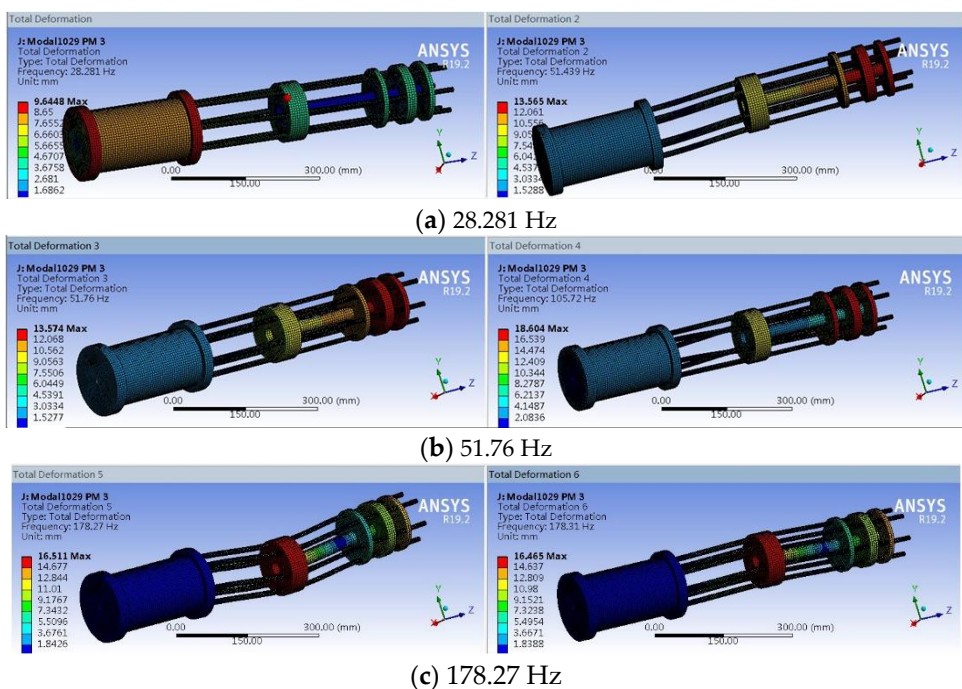

(**a**) 28.281 Hz

(**b**) 51.76 Hz

(**c**) 178.27 Hz

**Figure 12.** Prestress modal of M134. The M134 shoots at 2000 rpm, as shown in Table 2, which is 34.9 rad/s, and the intermediate state of the third and fourth-order. 34.9 Hz is from the third-frequency (28.281 Hz) to fourth-frequency (51.439 Hz). In addition, with the rpm of 3000, the barrels are neither twisted nor skewed but stretched along the axis of the barrel, which is very beneficial to shooting, with the fourth-frequency of 51.76 Hz, as Figure 12 shows.

## 5. Result Simulation

As we research the force of the rotary cantilever structure, it has to be researched in the ANSYS workbench. If the practical firing rate is 2000 rpm, then the rotation rate $\omega$ is 34.9 rad/s.

It is from the original parameters, such as elastic modulus E which is $2 \times 10^{11}$ Pa. maximum results are similar to Figures 13–16. In the plane perpendicular to the barrel axis, this deformation data is very small, and the data is very large in the root of the barrel. Additionally, in the simulation, add the bullet's x displacement as Figure 15 and the force on the bullet's as Figure 16, which is as Figure 8.

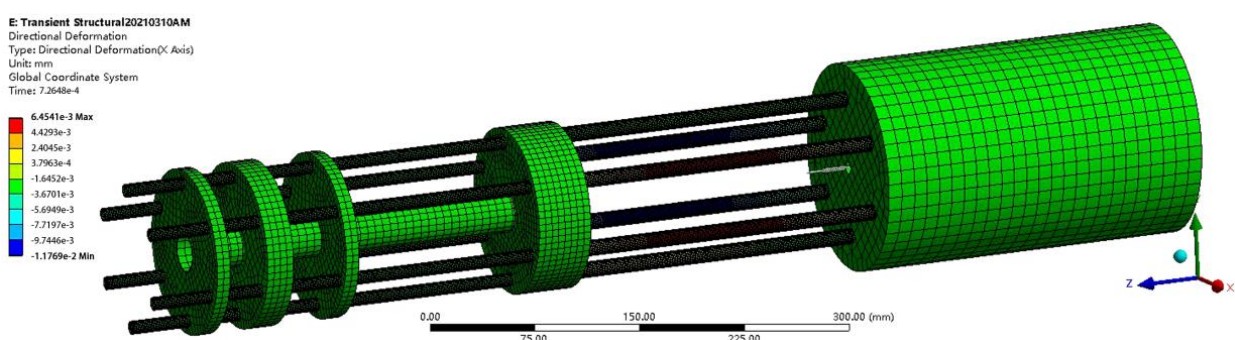

**Figure 13.** The X (transverse) displacement.

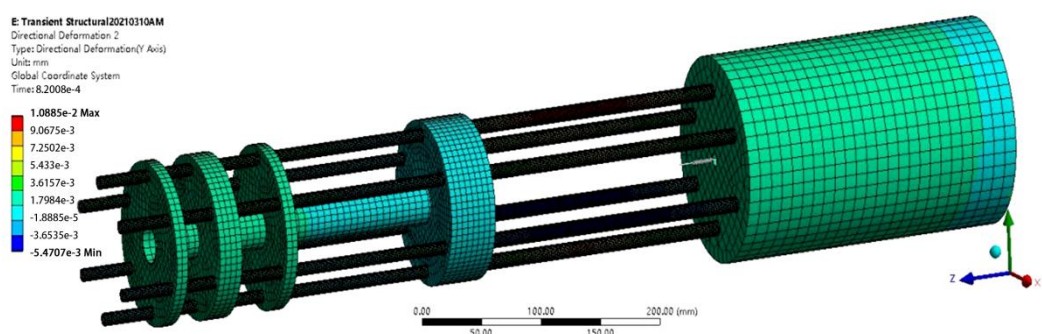

**Figure 14.** The Y (vertical) displacement.

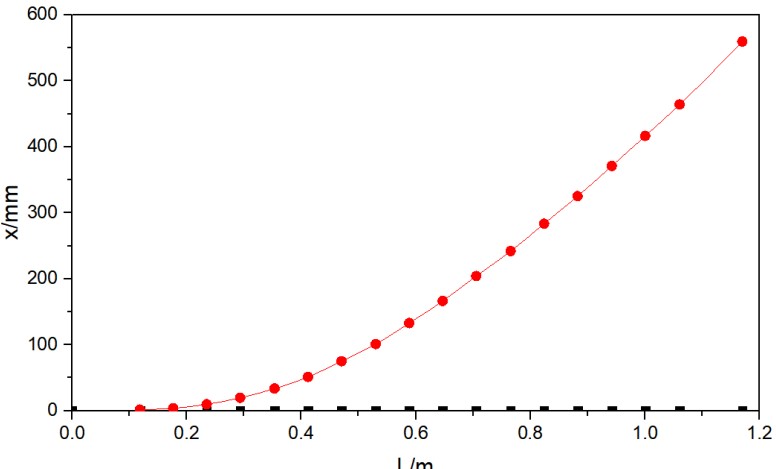

**Figure 15.** Bullet x (transverse) displacement.

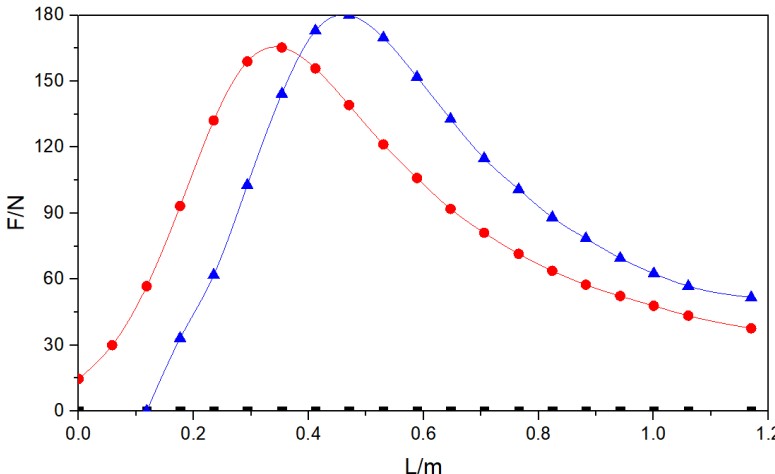

**Figure 16.** Bullet's force F1 and F2.

## 6. Optimization of the Cantilever Rotor System

### 6.1. Increase the Elastic Modulus of Barrel Material

The elastic modulus of the barrel material is at elastic modulus E which is $2 \times 10^{11}$ Pa. Let the elastic modulus by 10% of the upper and lower, the upper elastic modulus is $2.2 \times 10^{11}$ Pa and the lower elastic modulus is $1.8 \times 10^{11}$ Pa. In addition, the maximum of X and Y is shown in Figures 17 and 18.

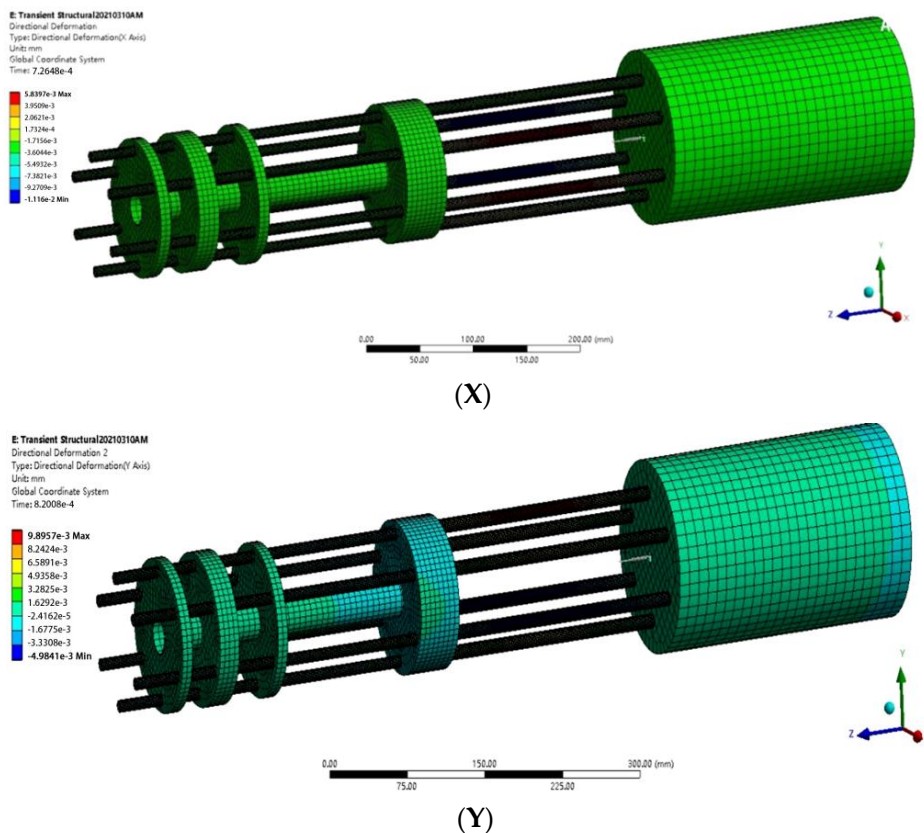

**Figure 17.** Maximum **X** (transverse) and **Y** (vertical) of elastic modulus of $2.2 \times 10^{11}$ Pa.

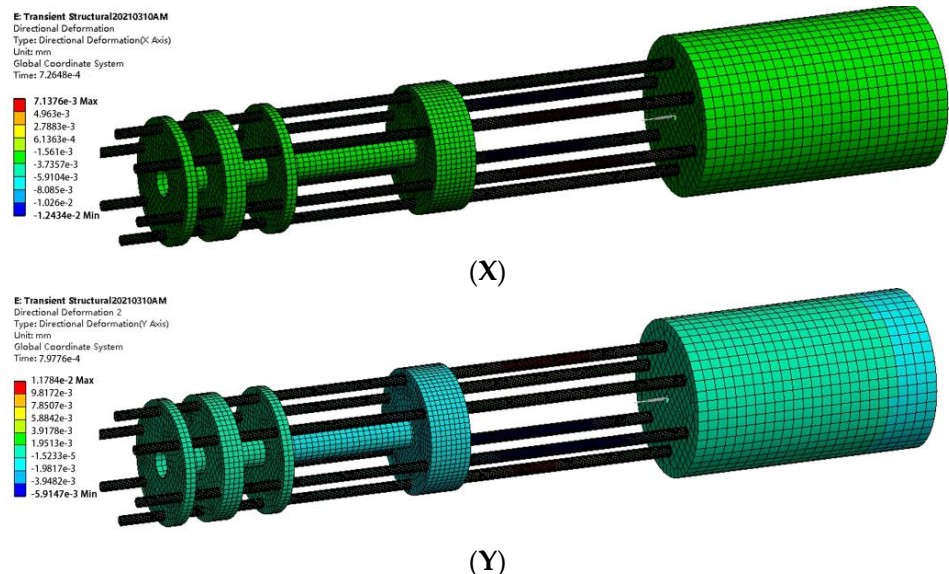

**Figure 18.** Maximum **X** (transverse) and **Y** (vertical) of elastic modulus of $1.8 \times 10^{11}$ Pa.

And the maximum of X and Y is shown in Table 4.

**Table 4.** Maximum of X (transverse) and Y about elastic modulus.

| Elastic Modulus/Pa | Maximum of X/mm | Maximum of Y/mm |
|---|---|---|
| $1.8 \times 10^{11}$ | 0.0071376 | 0.011784 |
| $2.0 \times 10^{11}$ | 0.0064541 | 0.010885 |
| $2.2 \times 10^{11}$ | 0.0058397 | 0.009857 |

The reductions are as follows:

$$\delta_x = \frac{0.0071376 - 0.0058397}{0.0071376} = 18.09\% \tag{19}$$

$$\delta_y = \frac{0.011784 - 0.009857}{0.011784} = 16.35\% \tag{20}$$

Therefore, with the increasing of the elastic modulus, the barrel deformation of X and Y decreases, reaching nearly 20%. The larger the E value of steel, the smaller the muzzle deformation.

### 6.2. Simulation by Changing the Position of the Second Enclosure

The second enclosure of M134 is 225 mm to the body parts of the Gatling gun, and the L value is 225 mm. We change the L value of the second enclosure, its L value is 200 mm or 250 mm. The second enclosure is closer to the weapon body if L is equal to 200 mm. In addition, the second enclosure is farther away from the weapon body if L is equal to 250 mm. The L is shown in Figure 19.

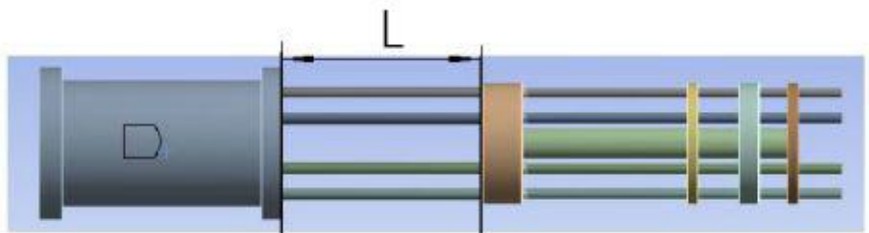

**Figure 19.** The distance from the second enclosure to the body part is L.

#### 6.2.1. The Weapon Frequency from L

The frequency from the L changes is from the Ansys Workbench, the frequency is the list in Table 5.

**Table 5.** L value brings the frequency of Weapon.

| L | Frequency | | | | | | | |
|---|---|---|---|---|---|---|---|---|
| | Fundamental | Second-Order | Third-Order | Fourth-Order | Fifth-Order | Sixth-Order | Seventh-Order | Eighth-Order |
| 225 | 19.477 | 19.658 | 61.735 | 61.911 | 72.52 | 166.31 | 200.43 | 200.63 |
| 200 | 20.018 | 20.215 | 56.419 | 58.168 | 65.615 | 165.17 | 166.63 | 178 |
| 250 | 18.489 | 18.664 | 54.895 | 55.108 | 63.652 | 165.89 | 219.77 | 220.01 |

From Table 5, the M134 can not fire at 1000 rpm, because this is close to the fundamental frequency.

As the frequency is shown in Figures 20–22. The L value is 225 mm (which is called the M134 prototype), the picture is as Figure 20.

If L value is 200 mm, which is as Figure 21.

Else if L value is 250 mm, which is as Figure 22.

By comparing the three schemes, we can see how appropriate the M134 is designed and far away from the firing frequency.

#### 6.2.2. The Barrel Deformation Closest to rpm

If the L varies from 200 mm to 250 mm, the performance of the M134 Gatling gun is shown in Figures 23–25 below.

From Figures 23–25, we can see that the frequency close to the launch frequency, the L value is 225 mm, which can only be regarded as the medium level. After all, the muzzle has a 1.5 mm offset, which is only about 1 mm offset from the 250 mm muzzle. Compared with 200 mm, the muzzle has an offset of nearly 6 mm.

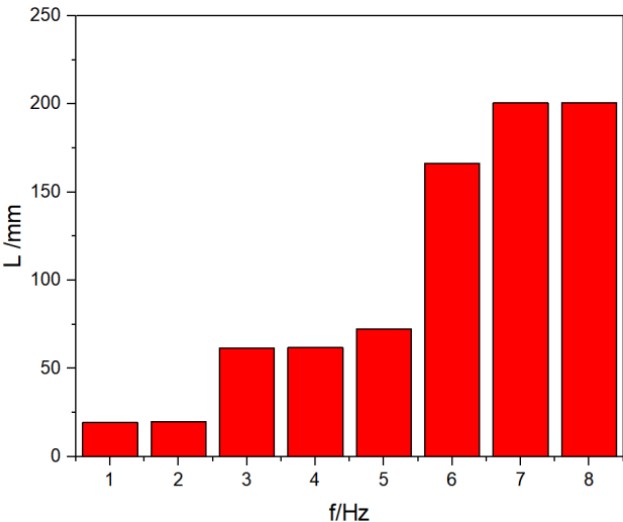

**Figure 20.** Frequency of L which value is 225 mm.

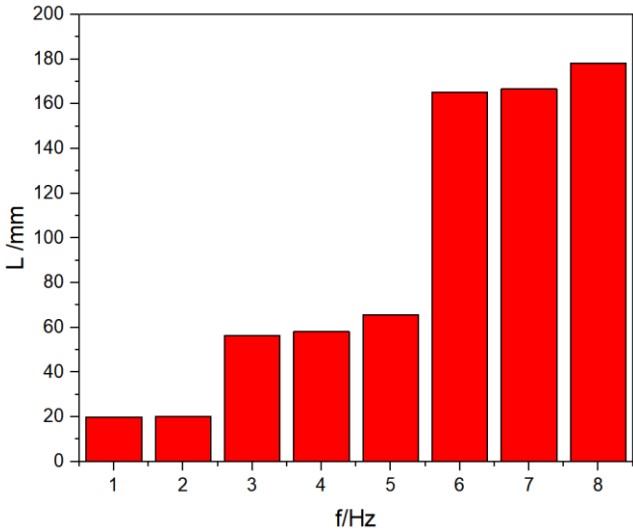

**Figure 21.** Frequency of L which value is 200 mm.

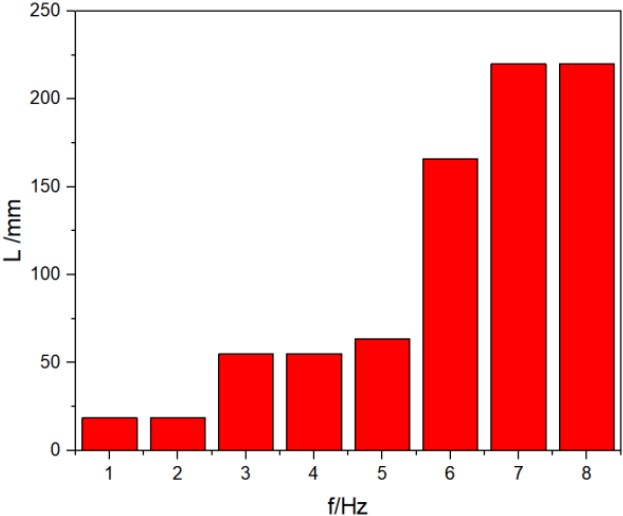

**Figure 22.** Frequency of L which value is 250 mm.

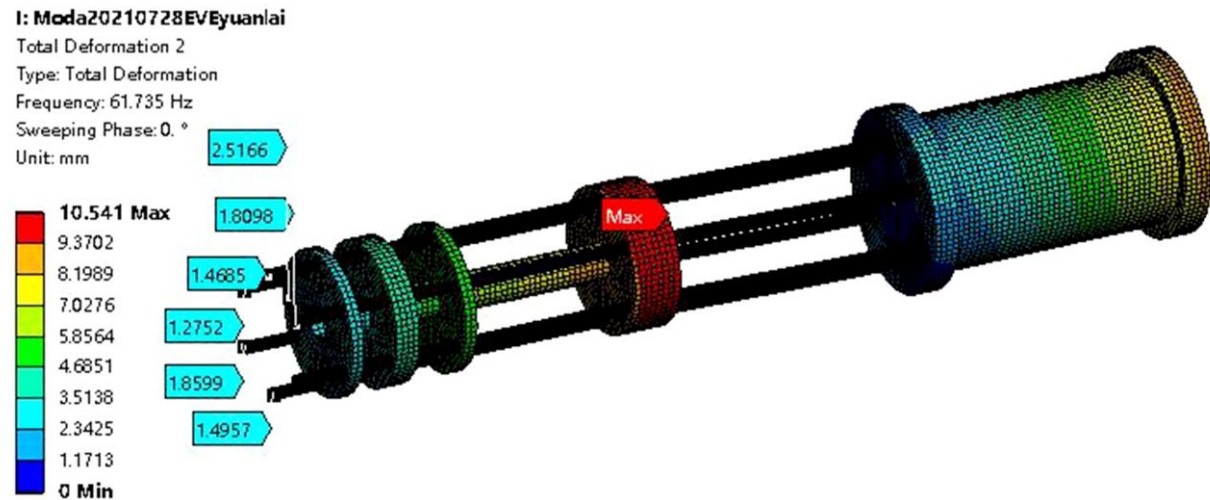

**Figure 23.** L value of 225 mm in M134.

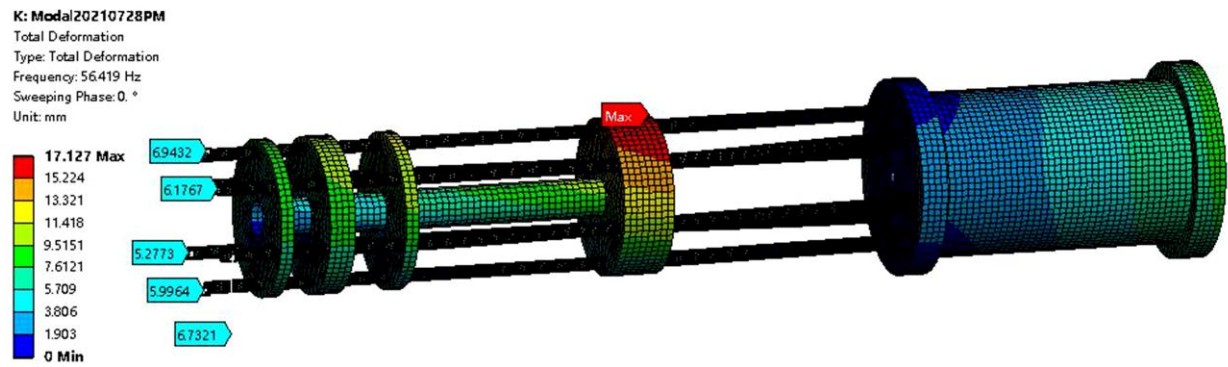

**Figure 24.** L value of 200 mm in M134.

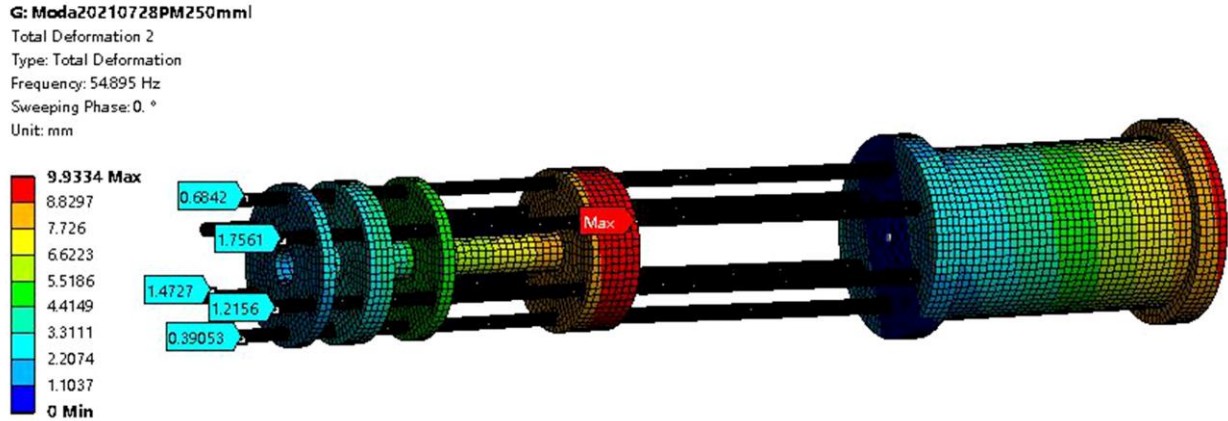

**Figure 25.** L value of 250 mm in M134.

*6.3. Simulation of Changing the Cross-Sectional Area*

That is not possible, because the cross-sectional area must be accurate. It is about 10% at most. Since it is a military product, it must be strictly required. If it reduces the effect by 10%, as shown in Figure 26.

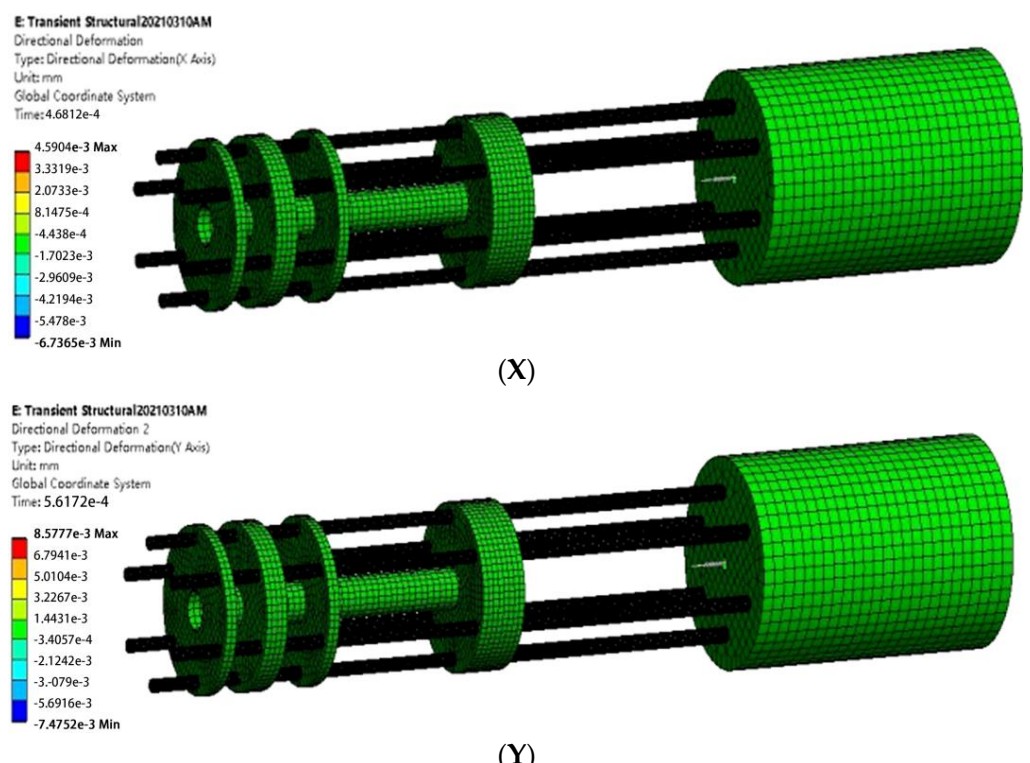

**Figure 26.** **X** (transverse) and **Y** (vertical) of the cross-sectional area reduce the effect by 10%.

And the maximum of X and Y is shown in Table 6.

**Table 6.** Maximum of X (transverse) and Y (vertical) about Cross-sectional area.

| Cross-Sectional Area | Maximum of X/mm | Maximum of Y/mm |
|---|---|---|
| Original | 0.0064541 | 0.010885 |
| Reduces the effect by 10% | 0.0045904 | 0.0085777 |

From Table 6, with Table 1, so the barrel is only 801.6 mm, and the original x result is 0.0060 mm, and 0.0108 mm in the Y direction, the new gun X direction is 0.0045 mm and Y direction is 0.0085 mm in. Now it is equivalent to the 600 m target, then the X offset of the original gun $x_0$ is 4.49 mm, the Y offset $y_0$ is 7.49 mm, and the X offset of the new gun $X_j$ is 3.37 mm and the Y offset $Y_j$ is 6.36 mm. Let's calculate as follows Equation (21):

$$\eta = \frac{\sqrt{x_j^2 + y_1^2}}{\sqrt{x_0^2 + y_0^2}} = \frac{\sqrt{3.37^2 + 6.36^2}}{\sqrt{4.49^2 + 7.49^2}} = 82.3\% \tag{21}$$

By reducing the cross-sectional area by 10%, a good result as Equation (21) can be achieved. The deformation of X and Y is smaller, but there are other components in it. If they can't be put down, they still have to be the same size as before [28].

## 7. Test and Data

We used the same number of barrels and the same configuration of the front structure of the barrel to shoot. Simulating shooting with 2000 rpm, and place the target 2000 m away. The shot image is shown in Figure 27.

Record the target center and the center position of the two shooting methods, and shown in Figure 28 below, the unit X and Y is meter. The absolute value decreases the range of R60 m.

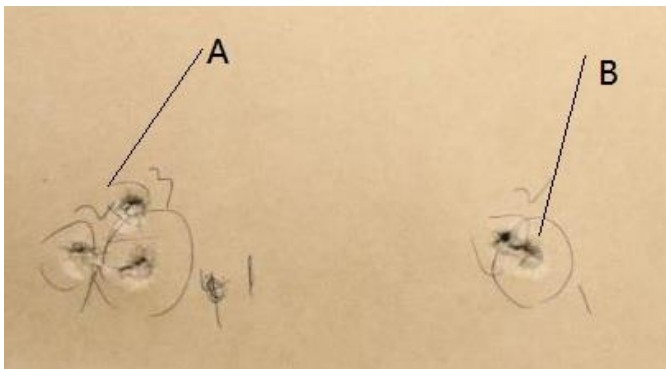

**Figure 27.** Shot image from Gatling gun. A—original Gatling gun; B—increase E Gatling gun.

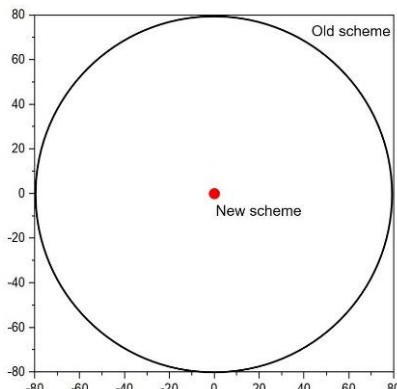

**Figure 28.** Two models calculation flying 2000 m.

## 8. Conclusions

After studying the M134 cantilever rotor, the following conclusions are obtained:

(1)　If the M134 Gatling gun rpm is 300–500, this M134 rotor is rigid, and for others, rpm is 1000–6000, the M134 rotor is the flexible rotor. It's still a good way to distinguish weapons according to the rpm. With the rpm of 3000, the barrels are neither twisted nor skewed, but stretched along the axis of the barrel, which is very beneficial to shooting, with the fourth frequency of 51.76 Hz. The best shooting is 300–500 for low-frequency shooting and 3000 for high-frequency shooting.

(2)　The increase of the elastic modulus is helpful to reduce the displacement of the barrel. Improve shooting accuracy according to the reduced vibration. The barrel deformation can be increased from $1.8 \times 10^{11}$ to $2.2 \times 10^{11}$. The X and Y directions of the muzzle can be reduced by 18%. The absolute value decreases the range of R60 cm from 2000 m away.

(3)　It is not by changing the cross-sectional area, although there are some good results, there are some other military products in it, the other reason is that the dimensions are fixed.

(4)　With the 3000 rpm, the barrels are neither twisted nor skewed, but stretched along the axis of the barrel, which is very beneficial to shooting, with the fourth-frequency of 51.76 Hz, the Gatling gun shoots at 3105.6 rpm (51.76 × 60). 3000 rpm is correct and reasonable.

**Author Contributions:** Data curation, Z.Y.; Formal analysis, J.X. and G.W.; Resources, Q.L.; Writing—original draft, H.Y. All authors have read and agreed to the published version of the manuscript.

**Funding:** National Natural Science Foundation of China #51175481.

**Informed Consent Statement:** Written informed consent has been obtained from the patient(s) to publish this paper.

**Conflicts of Interest:** The authors declared no potential conflict of interest in connecting to the research, authorship, and/or publication of this article.

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
