# Peer review of "Prestress Modal Analysis and Optimization of Cantilever Supported Rotor under the Unbalanced Axis Force and Moving Mass"

_applsci, doi:10.3390/app12104940_

Round 1

Reviewer 1 Report

Report on “Prestress Modal Analysis and Optimization of Cantilever Supported Rotor under the Unbalanced Axis Force and Moving Mass”.

The study by Hao Yang et al is an experimental work on the applied mechanics of the M134 Gatling weapon. The authors base their experimental study on the Bernoulli-Euler beam theory.

This is an extraordinary manuscript and the use of a Gatling weapon to study rotating vibrations with unbalanced axial forces and moving mass is original and interesting.

The main concern is the poor grammar in the manuscript; it has to be revised before the review process is done.

Additional comments include:

  • 1 is not written in an equation editor of any sort; it has to be corrected.
  • Why Eq. 2 is written in italic while Eq. 3 is not?
  • 4 appears to be smaller than Eq. 4.
  • 4 is not presented in an original format but was screenshotted, there is not enough data in the caption.
  • The same applies to Figures 5,6. These screenshots have been squeezed in the x-axis.
  • 5-8 appear in different formats as well.
  • Figures 7,8,9 does not include sufficient data in the caption. These figures are also screenshot-based. Original data must be provided.
  • Theoretical Model: The vertical deformation is represented by y1 and the longitudinal deformation is represented by y2, use, not y1.
  • Can you design Figure 11 in a more scientific format? Try to use PowerPoint.
  • Figure 12: is not accessible to the reader; it is squeezed on the x-axis.
  • Increase axis in Figures 16,17. In addition, where are the numbers on the L/m axis?
  • Figures 22,23 have been cut off, and crucial data is missing.
  • The fonts change in Figure 21.

Judging from the numerical data and the final result, the manuscript seems original.

However, poor editing and English grammar must be corrected before further review.

Reviewer 2 Report

This paper presents a method to implement the modal analysis and rotor optimization under unbalanced axis force and moving mass. The paper overall is interesting, and the results are relatively well presented. However, the writing in current form certainly needs to be improved. Below are the comments that authors can consider addressing to improve paper quality.

Abstract certainly needs to be rewritten. Instead of describing specific details, such as many numbers given, it is suggested to give a clear big picture of this research. What does this research try to address? What is the novelty of the research as compared to the state-of-the arts? What is key take-away from the results obtained.

Introduction is too short to involve the necessary discussion. It is not good to just list other’s works without tight connection. The most important thing is to highlight your proposed method based upon the literature review. In other words, why do you use the proposed method? However, such information cannot be found in current Introduction.

In Table 2, authors mentioned the rotation rate, which is an important factor for subsequent analysis. What is the exact physical meaning of rotation rate? Is it equal to rotor speed in Table 2? It is very confusing about how to find the rotation rate information in Table 2.

One keyword of this paper is “optimization”. How did authors carry out optimization? It seems there is lack of associated details. Just use trial-and-error attempts or use rigorous optimization method?

Figure 11, consider making the coordinates clearer, i.e., X, Y, Z.

Figure 12, 17,21,22,23, the labels are too small to read.

For some very short paragraphs, it is suggested to merge them.

The paper needs to be proofread carefully to correct the grammar issues and other improper notations intensively, such as

“Study it and draw a conclusion, which is appropriate for us to study weapons and understand the reason for weapon vibration” correct the grammar error.

“How to judge whether the Gatling gun is rigid or flexible when firing, and how to judge the muzzle vibration to improve the killing effect of the Gatling gun.” What does that mean? Raise the question to audience? No further explanation/discussion is given. Authors should completely avoid the academic writing like this.

Indices of Equation (6) (7) (8) are not supposed to be “italic”

“And the maximum of X and Y is shown in Figure 18 and Figure 19.”  Authors should explicitly point out what are X and Y. Maximum values of X and Y displacements? For the figure captions, it is a bad idea to put (X) and (Y) there. It is supposed to be “(a). maximum value of X displacement ” or something similar…

The formats of symbols in equations are inconsistent. Some are regular, and some are italic…

Round 2

Reviewer 1 Report

I have read the manuscript and the attached response.

The authors have made significant progress to improve the readability of the manuscript.

Some matters still must be addressed:

  1. Figure 5,6 are still squeezed in the x axis; it must be fixed.
  2. I can't understand what exactly is shown in Figure 9. I suggest labelling the left side as 9(a), right side as 9(b) and explain precisely what is drawn there.
  3. Figure 10 still looks very poor. The hand drawing in paint is not acceptable in a scientific publication. I also suggest splitting it to 10(a) and 10(b) and explain what is shown in the caption.
  4. Please give more details in the caption of figure 19.
  5. It seems like Figure 28 is still presented in a screenshot format, it is suggested to improve it.

Author Response

请参阅附件

Reviewer 2 Report

Careful proofread and respective correction still are needed. For example, authors used both the fist capital letter of "Figure xx" and first lower case letter of "figure XX".
